# The Role of FBXW7 in Gynecologic Malignancies

**DOI:** 10.3390/cells12101415

**Published:** 2023-05-17

**Authors:** Riccardo Di Fiore, Sherif Suleiman, Rosa Drago-Ferrante, Yashwanth Subbannayya, Sarah Suleiman, Mariela Vasileva-Slaveva, Angel Yordanov, Francesca Pentimalli, Antonio Giordano, Jean Calleja-Agius

**Affiliations:** 1Department of Anatomy, Faculty of Medicine and Surgery, University of Malta, MSD 2080 Msida, Malta; 2Sbarro Institute for Cancer Research and Molecular Medicine, Center for Biotechnology, College of Science and Technology, Temple University, Philadelphia, PA 19122, USA; 3BioDNA Laboratories, Malta Life Sciences Park, SGN 3000 San Gwann, Malta; 4School of Biosciences, Faculty of Health and Medical Sciences, University of Surrey, Guildford GU2 7XH, UK; 5Whipps Cross Hospital, Barts Health NHS Trust, Leytonstone, London E11 1NR, UK; 6Department of Breast Surgery, “Dr. Shterev” Hospital, 1330 Sofia, Bulgaria; 7Research Institute, Medical University Pleven, 5800 Pleven, Bulgaria; 8Bulgarian Breast and Gynecological Cancer Association, 1784 Sofia, Bulgaria; 9Department of Gynecological Oncology, Medical University Pleven, 5800 Pleven, Bulgaria; 10Department of Medicine and Surgery, LUM University “Giuseppe DeGennaro”, 70010 Casamassima, Italy; 11Department of Medical Biotechnologies, University of Siena, 53100 Siena, Italy

**Keywords:** FBXW7, ubiquitin-proteasome system, gynecologic cancers, epigenetic, mutations, miRNAs, LncRNAs, target therapy

## Abstract

The F-Box and WD Repeat Domain Containing 7 (FBXW7) protein has been shown to regulate cellular growth and act as a tumor suppressor. This protein, also known as FBW7, hCDC4, SEL10 or hAGO, is encoded by the gene FBXW7. It is a crucial component of the Skp1-Cullin1-F-box (SCF) complex, which is a ubiquitin ligase. This complex aids in the degradation of many oncoproteins, such as cyclin E, c-JUN, c-MYC, NOTCH, and MCL1, via the ubiquitin-proteasome system (UPS). The FBXW7 gene is commonly mutated or deleted in numerous types of cancer, including gynecologic cancers (GCs). Such FBXW7 mutations are linked to a poor prognosis due to increased treatment resistance. Hence, detection of the FBXW7 mutation may possibly be an appropriate diagnostic and prognostic biomarker that plays a central role in determining suitable individualized management. Recent studies also suggest that, under specific circumstances, FBXW7 may act as an oncogene. There is mounting evidence indicating that the aberrant expression of FBXW7 is involved in the development of GCs. The aim of this review is to give an update on the role of FBXW7 as a potential biomarker and also as a therapeutic target for novel treatments, particularly in the management of GCs.

## 1. Introduction

Protein degradation is a crucial process for various cellular processes, such as cell growth, cell cycle, differentiation, and apoptosis [1,2]. Dysregulation of protein degradation machinery can result in aberrant protein stabilization, leading to oncogenesis due to the accumulation of oncogenic proteins. The ubiquitin-proteasome system (UPS) is the primary regulatory pathway leading to protein degradation in eukaryotic cells [3]. The UPS comprises E1, a ubiquitin-activating enzyme; E2, a ubiquitin-conjugating enzyme; and E3, a ubiquitin ligase. The activation of ubiquitin by E1 is followed by the transfer of ubiquitin to E2 via a thioester exchange reaction. In the end, ubiquitin is transferred from E2 to a substrate using E3, leading to the substrate’s degradation by the 26S proteasomes. Modulation of E3 ubiquitin ligase function has been demonstrated to be a significant factor in cancer initiation and progression [4,5]. Among the different types of E3 ubiquitin ligases, the Skp1-Cullin1-F-box (SCF) complex is the most well-studied. The SCF complex comprises Cul1, F-box protein, Rbx1, and Skp1 [5]. The catalytic core of SCF is the Cul1 domain, whereas the Skp1 domain adds the F-box to the Cul1 and Rbx1 domains, which is critical for the SCF complex’s catalytic function [6,7]. The F-box domain acts as a substrate receptor and can recruit substrates for subsequent ubiquitination [5]. F-box proteins behave as substrate recognition subunits that confer specificity to individual SCF-type ligases. An F-box domain within the NH2-terminal region of each F-box protein can bind to the SKP1-CUL1 complex, while there are numerous protein-interaction domains in the COOH-terminal region which are responsible for substrate recognition. F-box proteins are classified into three types based on their different COOH-terminal regions: FBXL (containing leucine-rich–repeat domains), FBXW (containing WD40-repeat domains), and FBXO (containing other protein-interaction domains or a non-recognizable domain). Among approximately 70 F-box proteins that have been identified in humans, FBXW7 (also known as FBW7, hCDC4, hAGO, or Sel10) has the highest frequency of mutations in cancer [1,8,9,10,11], indicating its crucial role as a tumor suppressor in oncopathogenesis. Most FBXW7 substrates are proto-oncoproteins, including c-JUN, cyclin E, c-MYC, MCL1, and NOTCH [9]. However, recent studies have identified novel oncogenic FBXW7 mutations in human T-cell leukemia virus (HTLV-I) transformed adult T-cell leukemia (ATL) cells, suggesting a possible carcinogenic role for FBXW7 [12]. Co-expression of FBXW7 mutations D510E and D527G with either HTLV-I’s oncoprotein Tax, mutated c-Myc (F138C) or mutated p53 (R276H) leads to transforming activity [12]. Moreover, this activity promotes IL-2-independent growth of Tax-immortalized human T cells and increases the formation of tumors in a xenograft mouse model of ATL [12]. These findings demonstrate that FBXW7, which typically functions as a tumor suppressor, may also act as an oncogene under certain circumstances. Therefore, the aim of this review is to provide an update on the role of FBXW7 as a potential biomarker and therapeutic target for novel oncological treatments, including GCs.

## 2. FBW7: Structure and Its Substrates

The FBXW7 gene is situated in chromosome 4q31.3 and contains 13 coding exons and 4 untranslated introns. It encodes three distinct isoforms: FBXW7-α, FBXW7-β, and FBXW7-γ, produced through alternative splicing of the same transcript. These isoforms are localized predominantly in the nucleoplasm, cytoplasm, and nucleolus, respectively [8,11]. All of the isoforms share conserved domains at the C-terminal region and differ only in their N-terminal region. Each isoform contains an F-box domain, a dimerization domain (DD), and seven tandem WD40 repeats (Figure 1A). The F-box domain binds the Skp1 component of the SCF complex, while the DD helps to bind the substrate [13]. The WD40 repeat forms a propeller that recognizes the phosphorylated substrate via the conserved phosphorylated domain, Cdc4 phosphodegron (CPD), which is phosphorylated by glycogen synthase kinase 3 (GSK3) [14,15].

As already mentioned, FBXW7 is one of the main components of the SCF complex. It helps in the recognition of substrates for ubiquitination and eventually proteasomal degradation by the 26S proteasome [14] (Figure 1B). Under normal biological conditions, FBXW7 ensures the maturation of bone marrow erythroid cells via the regulation of cyclin E expression [16]. It is also involved in the differentiation and proliferation of both stem and progenitor cells [17]. FBXW7 has also been shown to be important for the pluripotency of embryonic stem cells (ESCs) as it controls the stability of the c-Myc protein [14]. Above all, FBXW7 is regarded as a strong p53-dependent tumor suppressor controlling human cell growth, cell cycle progression and tumor development through the direction of various oncoproteins such as Aurora A, cyclin E, c-Jun, c-Myb, c-Myc, JUNB, KLF5, Mcl1, MED13, mammalian target of rapamycin (mTOR), NF1, NFκB2, Notch, NRF1, p63, SREBP, NONO, and for ubiquitin-mediated proteolysis [18,19,20,21,22,23,24,25].

FBXW7 can function both as a monomer or as a dimer. It is thought that this may affect the selection of the target and binding strength. Dimerization of FBXW7 may also play a crucial role in cancer cells where there is one copy of mutated *FBXW7* allele. In human cancers, *FBXW7* is often inactivated via genetic and epigenetic mechanisms and post-transcriptional modifications [1,8,9,10,11]. The loss of FBXW7 has been shown to be also strongly associated with carcinogenesis, tumor metastasis, and resistance to chemo-, radiation-, and immuno-therapies, leading to poorer outcomes [2,26].

## 3. Regulation of FBXW7

To date, the majority of studies focus on the discovery of the ubiquitin targets of the FBXW7 ubiquitin ligase pathway. However, it is still unclear how FBXW7 itself is regulated in different human cancers. To this end, emerging evidence has demonstrated that numerous molecules such as CCAAT/enhancer-binding protein-δ (C/EBP-δ) [27], p53 [28], EBP2 [29], Hairy and Enhancer-of-split homologues 5 (Hes-5) [30], Numb-4 [31], Peptidyl-prolyl cis-trans isomerase NIMA-interacting 1 (Pin1) [32], as well as microRNAs (miRNAs) (Table 1) including miR-25 [33], miR-27a [34], miR-92a [35], miR-129-5p [36], miR-182 [37], miR-223 [38], and miR-503 [37], have all been found to regulate the expression of *FBW7*. In addition, several long noncoding RNAs (lncRNAs) inhibit miRNA activity by acting as miRNA “sponges” (Table 1).

The lncRNA c-MYC inhibitory factor (MIF) enhances the negative effect of miR-586 on the abundance of FBXW7, leading to increased degradation of c-MYC [39]. Upregulation of lncRNA-MIF transcription has been demonstrated to be induced by c-MYC, indicating a feedback loop between lncRNA-MIF and c-MYC that finely controls the amount of c-MYC. Moreover, other lncRNAs, including TINCR, CASC2, MALAT1, and MT1JP, have been reported to function as miRNA sponges that can prevent miRNA-mediated inhibition of FBXW7 expression via miR-367, miR-544a, miR-155, and miR-92a-3p, respectively [40,41,42,43]. Figure 2 illustrates the upstream regulatory mechanisms and downstream substrates of FBXW7.

## 4. Post-Translational Regulation of FBXW7

Post-translational modifications of FBXW7 involve its phosphorylation, autoubiquitination, deubiquitination, dimerization, and localization [44].

**Phosphorylation.** FBXW7 function is regulated by multiple kinases that catalyze its phosphorylation [44,45]. The Thr205 phosphorylation of FBXW7α, leading to its ubiquitylation and proteasomal degradation, is directly mediated by extracellular signal-regulated kinase (ERK) [46]. Additionally, Polo-like kinase 1 (PLK1) destabilizes FBXW7 by phosphorylation at Thr284 and Ser58 in FBXW7γ, which is equivalent to Thr402 and Ser176 in FBXW7α [47], and PLK2 destabilizes it by phosphorylating Ser176, and to some extent, Ser25 and Ser349 in FBXW7α [48]. However, FBXW7α is stabilized by phosphorylation at Ser227 by serum and glucocorticoid-regulated kinase 1 (SGK1) or phosphoinositide 3-kinase (PI3K), which inhibits autocatalytic ubiquitin transfer [49,50]. Moreover, Ser10 and Ser18 are phosphorylated in a protein kinase C–dependent manner within the isoform-specific NH2-terminal region of FBXW7α, with phosphorylation at Ser10 shown to prevent its nuclear localization [51].

**Autoubiquitination.** FBXW7’s regulation extends beyond its role in ubiquitination and degradation, as it can also be subject to autoubiquitination. Peptidyl-prolyl cis-trans isomerase NIMA interacting 1 (Pin1) decreases FBXW7 dimerization, promoting its destabilization and self-ubiquitination [32]. Additionally, FBXW7 stability is influenced by SCF-dependent mechanisms. One example is COP9 signalosome complex subunit 6, which is a member of the COP9 signalosome complex, which enhances FBXW7 autoubiquitination and subsequent proteasome-mediated degradation via the regulation of Cul1 neddylation [52].

**Deubiquitination.** USP28, a deubiquitinating enzyme, has been found to regulate FBXW7 activity [45]. Remarkably, the removal of one copy of USP28 preserves stable FBXW7 and promotes the degradation of its substrate [45]. Conversely, complete USP28 knockout leads to FBXW7 degradation, resulting in the accumulation of FBXW7 substrates, while the overexpression of USP28 stabilizes both FBXW7 and its substrates. As a result, both the absence and excess of USP28 contribute to Ras-driven oncogenic transformation. This dual control of FBXW7 activity by USP28 is believed to be a protective mechanism that maintains healthy levels of proto-oncogenic FBXW7 substrates. However, this equilibrium is disrupted by USP28 loss or overexpression [45].

**Dimerization.** FBXW7, through a conserved D domain, is capable of forming dimers. However, mutations in endogenous FBXW7 have been demonstrated to prevent dimer formation [53]. Dimerization aids in the ubiquitination of FBXW7 substrates with low-affinity degrons but not those with high-affinity degrons, for example, cyclin E and Myc [13,53]. Additionally, Pin1-mediated isomerization and human FBXW7 phosphorylation at Ser205 increase its autoubiquitination by inhibiting dimerization [32].

**Localization.** The deviant localization of FBXW7 interferes with its association with substrates. An example of this is evident in acute myelogenous leukemia, where nucleophosmin mutations, necessary for the nucleolar localization of FBX7γ, lead to FBX7γ instability, consequently increasing c-Myc expression [54]. Furthermore, phosphorylation of FBXW7α at Ser10 hinders one of its nuclear localization signals [51].

## 5. Genetic and Epigenetic Alterations Cause *FBXW7*-Inactivation

FBXW7 inactivation, resulting from mutations, deletions, or epigenetic modifications, is a major contributor to cancer progression and metastasis [55,56]. Monoallelic or biallelic deletions or promoter hypermethylation of the FBXW7 gene are frequently observed in various malignancies, such as breast, bladder, cervical, lung, esophagus, stomach, liver, and pancreas cancer [57]. Missense mutations affecting the critical arginine residues in the β-propeller’s phosphate-binding pockets are also common [58]. These mutants are believed to act as dominant negative alleles and eventually cause functional inactivation of the wild-type protein [57,58]. Although tumors typically express a functional wild-type protein by retaining the second wild-type allele of FBXW7, in mouse models, monoallelic deletions display a milder tumor phenotype than when there is complete gene loss [59,60].

FBXW7α has a broad tissue distribution and is ubiquitously expressed, as demonstrated by numerous in vitro, in vivo, and clinical studies [1]. However, FBXW7β demonstrates differential expression in diverse cell lines and tissues, with its promoter being epigenetically regulated through histone and DNA modifications. In fact, up to 51% of breast cancer tumors and 43% of other cancer cell lines have been found to have methylated FBXW7β promoters [55]. The hypermethylation of the FBXW7 promoter is usually associated with p53 mutations, which upregulate DNA methyltransferase 1 (DNMT1) and suppress FBXW7 expression. Moreover, histone modifications play a crucial role in regulating FBXW7 expression. For example, EZH2, a histone methyltransferase, can silence FBXW7 by adding three methyl groups to the histone H3 residue, H3K27me3 [61]. Notch signaling is another important regulator of FBXW7 expression. The upregulation of the Notch target gene and Hes5 transcriptional repressor can suppress FBXW7 gene expression and lead to a positive feedback loop which enhances the FBXW7 loss-of-function phenotype [62].

In the turnover of substrate, the effects of FBXW7 mutations are mainly context-dependent, with the frequently tested FBXW7 substrate level remaining unaffected in other tissues of FBXW7Mut/+ mice, except for TGIF1 and KLF5 [63]. FBXW7 mutation can also ameliorate Apcmin-driven intestinal tumor growth. However, the adenomas arising in these mice still possess normal levels of Jun, Myc, and Notch. Therefore, it is likely that heterozygous FBXW7 mutations can promote tumorigenesis by regulating “non-canonical” substrates, such as KLF5 and TGIF1 [58]. Given FBXW7′s critical role in maintaining physiological substrate levels, understanding the mechanisms that control its activity is essential.

## 6. Deregulation of FBXW7 in Gynecologic Cancers

GCs, which include ovarian, uterine/endometrial, cervical, vaginal, and vulvar cancers, cause a huge worldwide health-socio-economic burden due to their high incidence and mortality among women, irrespective of age [64,65]. Lack of screening, limited awareness of specific symptoms, late diagnosis, or even misdiagnosis, combined with limited treatment options for advanced GCs, are the main contributing factors leading to the high morbidity and mortality, thus stressing the need for further advances in the area of GC [65].

Being a tumor suppressor, *FBXW7* is the gene which is most commonly mutated among all the genes encoding F-box proteins in malignancies in humans [11,66]. In the meta-analyses of the cBioPortal Database [56,67] that we have performed, we found an overall *FBXW7* somatic mutation rate of 13% in GCs (255 cases out of 1958 tested; Appendix A), although different GCs were exhibiting different mutational spectra (Figure 3A). Most of the FBXW7 mutations are single nucleotide changes, leading to single amino acid substitutions within the WD40 domains that are responsible for binding substrates. The mutations of these key residues will prevent FBXW7 from binding with its oncogenic substrates. In Figure 3B, three mutation hotspots, R465, R479, and R505, are shown. These represent up to 40.7% of mutations (110/270) found in all FBXW7 mutations. Now we will proceed by describing the main types of GCs, including some rare types, and the latest findings on the role of FBXW7 in these particular tumors. The possible clinical significance of FBXW7 in GCs will also be outlined. 

### 6.1. FBXW7 in Ovarian Cancer

Ovarian cancer (OC) is a heterogeneous tumor with different pathophysiological development and clinical management and outcome. OC accounts for 5% of all malignancies in women [68]. Based on histopathological, immunohistochemical, and molecular genetic analysis, ovarian carcinoma is classified into five types: high-grade serous (the commonest type at 70%); clear cell (10%); endometrioid (10%); mucinous 3%; low-grade serous (<5%) (WHO 2020), and even rarer, Brenner tumor [69,70]. Different epithelial malignancies have different origins and morphologies with different biological behavior [71]. Mucinous carcinoma arises from germ cells; low-grade serous carcinoma arises from the fallopian tube; endometrioid, clear cell, and seromucinous carcinomas arise from the endometrium, while malignant Brenner tumors develop from the transitional epithelium. The majority of these carcinomas develop progressively from benign and borderline precursor lesions to malignant tumors [71,72]. Most of these OCs tend to be genetically stable, with mutations in different genes such as BRAF, CTNNB1, KRAS, and PTEN. However, in high-grade ovarian serous carcinoma, there is a lot of genetic instability, which is characterized by the loss of BRCA1-2 and TP53 mutations. It is a highly aggressive neoplasm, typically spreading to the omentum and mesentery and accompanied by ascites [71,72]. To date, there is a lack of effective clinical screening tools for OC, with approximately 70% of cases being diagnosed at an advanced stage [73]. In the United States alone, around 21,000 new cases of OC are diagnosed annually, with a mortality rate of 62% and a low five-year survival rate of only 20–30% [68]. Therefore, there is an urgent need for the introduction of highly sensitive and specific diagnostic tools to identify OC at an earlier stage and the development of new therapeutic approaches to improve the overall survival rates.

In OC, there is evidence of inactivation and functional loss of FBXW7, resulting from complex genetic and epigenetic alterations such as deletion, somatic mutation, and hypermethylation (Table 2) [74]. The frequency of the FBXW7 gene mutation is reported to be approximately 2.5 ~ 8.3% [21,75]. Boyd et al. found that the FBXW7 and KIAA1462 genes were mutated in serous borderline tumor (SBT) of the ovary [76]. Ovarian SBT is a unique histopathologic entity that is thought to be an intermediate between invasive low-grade serous carcinoma and benign cystadenoma of the ovary. These findings suggest that these mutations are novel candidates for the pathogenesis of ovarian SBT [76]. In addition, low or absent FBXW7 expression was commonly present in 19q12 amplified/high cyclin E1 cases of high-grade serous OC [77]. FBXW7 is significantly decreased in OC, and this has been associated with the DNA methylation status of the 5′-upstream regions of FBXW7 and p53 mutations [74].

The FBXW7 gene has been identified as a tumor suppressor in ovarian cancer (OC), as reported in studies [74,78]. The expression of FBXW7 is especially important in high-grade serous OC cells, as it plays a significant role in the sensitivity of these cells to anti-tubulin chemotherapeutic drugs [79]. A study conducted by Xu et al. focused on the expression of FBXW7 protein-coding transcript isoforms (α, β, and γ) in ovarian serous cystadenocarcinoma. Their research showed that FBXW7γ acts as a tumor suppressor and might be the only FBXW7 transcript that is related to prognosis in this particular type of cancer [80].

In both OC cell lines and tissues, Circ-BNC2 downregulation has been linked to a higher FIGO stage and lymph node metastasis. On the other hand, overexpression of Circ-BNC2 led to an upregulation of FBXW7 through sponging miR-223-3p, resulting in a reduced proliferation, migration, and invasion of OC cells [81]. Another study revealed that FBW7 also inhibits the development of OC by targeting the N6-methyladenosine binding protein YTHDF2 [82]. In addition, Miao et al. found that Titin-antisense RNA1 (TTN-AS1) is downregulated in both OC cells and tissues and has a positive correlation with FBXW7 expression. TTN-AS1 regulates FBXW7 expression by modulating miR-15b-5p, exerting a tumor-suppressive role in the development of OC. This suggests that the TTN-AS1/miR-15b-5p/FBXW7 axis could serve as a potential therapeutic biomarker for OC [83]. Astragalus polysaccharide (APS), a natural antioxidant present in Astragalus membranaceus, has been shown to suppress OC cell growth in vitro via the miR-27a/FBXW7 axis. This highlights the therapeutic potential of APS in OC treatment [84]. However, the exact role of FBW7 in OC progression remains inadequately understood and requires further exploration.

### 6.2. FBXW7 in Cervical Cancer

According to research, cervical cancer (CC) is the second leading cause of tumor-related death in women globally [85], with persistent infection of “high-risk” human papillomaviruses (HPVs), particularly HPV16 and HPV18, being the primary cause [86,87]. Other risk factors that have been linked to CC include early sexual activity [88], multiple sexual partners [89], infections such as HIV, herpes simplex virus type II and chlamydia [90], genetic factors similar to active oncogenes such as ATAD2, PIK3CA, and CRNDE, and tumor-suppressor genes such as RASSF1A, TP53, and NOL7 [91], and smoking [92]. Although Papanicolaou smears and liquid-based cytology have been the traditional screening methods for pre-invasive cervical disease [93], primary HPV screening is now being prioritized. Despite improvements in screening, detection, and treatment methods, including surgery, radiotherapy, and chemotherapy, early lymph node metastasis can still result in a poor prognosis for CC patients, with a five-year survival rate of approximately 40% [94,95]. More research is needed to identify potential molecular therapeutic targets that could improve the management of patients with advanced or recurrent CC.

Studies have shown that the FBXW7 gene plays a role in CC (Table 3). The mutation frequency of the FBXW7 gene is approximately 1.5% to 15% [96,97,98,99,100,101,102], with some mutations, such as R465C, R479Q, and R505G, also observed [103]. Alterations in FBXW7 and PIK3CA are believed to be the earliest changes that trigger malignant progression [104]. Five non-synonymous mutant genes, including FBXW7, were found in metastatic relapse significantly mutated (MSG) genes among MRCC samples [105]. Patients with any detectable MSG mutations had shorter progression-free and overall survival times than those without detectable MSG mutations [105]. Furthermore, the immune subtype of cervical squamous cell carcinoma (CSCC) HPV16-IMM, which has mesenchymal features and a strong immune response, had significant mutations at FBXW7 and epigenetic silencing [106]. Two FBXW7 mutations, R479P and L443H, were found to promote cell proliferation, migration, and invasion in CC cells in a study involving 145 CSCCs [107]. In another study using TCGA-CESC and GSE44001 datasets, 218 out of 291 patients (74.91%) with oxidative stress-related gene mutations were investigated, with FBXW7 mutations accounting for 12% of them [108]. A five oxidative stress-related gene signature was created to predict overall survival, and three subgroups based on genes linked to oxidative stress were generated to guide individualized therapy for CSCC patients. Preclinical evidence suggests that oxidative stress-related subtypes and Risk Score may be useful for the precisely tailored treatment of CSCC patients [108].

In CC, miR-92a is significantly upregulated and binds to the 3′UTR of FBXW7, inhibiting its expression and promoting tumor progression and invasion [35]. Xu et al. found that low FBXW7 expression was associated with high histologic grade, lymphovascular space invasion, and metastasis, and patients with low FBXW7 expression had poor progression-free and overall survival [109]. Downregulation of the long intergenic nonprotein-coding RNA 173 (LINC00173) in CC tissues is correlated with poor survival, and LINC00173 overexpression increases FBXW7 levels by regulating miR-182-5p, suppressing the proliferation and invasion of CC cells [110]. Additionally, miR-27a-3p and miR-103a-3p have also been implicated in CC development and progression. miR-27a-3p targets FBXW7, leading to its downregulation and promoting tumor growth [111]. miR-103a-3p is significantly upregulated in CC tissues and correlates with more aggressive histological grades, higher FIGO stage, and distant metastasis, leading to poor overall survival. miR-103a-3p also targets FBXW7, suggesting that it functions as an oncogene in CC by inhibiting FBXW7 [112]. These findings highlight the important role of FBXW7 in inhibiting CC development and progression and the potential therapeutic significance of targeting the miRNA-FBXW7 axis.

### 6.3. FBXW7 in Endometrial Cancer

Endometrial cancer (EC) is the most frequently diagnosed GC in developed countries [113]. Risk factors include hyperestrogenism due to early menarche, obesity, nulliparity, and late menopause. Advanced age, diabetes mellitus, Lynch syndrome, breast cancer, tamoxifen therapy, and radiotherapy have also been associated with EC [114]. Gene mutations are being used for EC classification [115]. EC can be divided into endometrioid (Type I), which occurs in around 80% of patients, and non-endometrioid (Type II) in the rest [116,117]. Non-endometrioid ECs include endometrial serous carcinoma, clear-cell carcinoma, and carcinosarcoma. Type 1 ECs have alterations in several different genes, including CTNNB1, KRAS, PTEN, and DNA characterized by microsatellite instability (MSI) [118,119]. In contrast, Type 2 EC tumors are defined as having TP53 mutations, increased CDH1 expression, amplification of HER2, and a high Ki-67 (MIB1) score, which is a marker of proliferation. The standard treatment is surgical, involving total hysterectomy and bilateral salpingo-oophorectomy, which is typically effective in the case of stage I disease [120]. However, when EC is advanced, surgery needs to be followed by radio- and/or chemotherapy. Despite advances in management for EC, survival rates have not improved significantly. Thus, improving the ability of identifying the risk factors and formulating novel management plans are essential for improving the prognosis and survival rate of patients with EC [121].

FBXW7 has been shown to be mutated in EC (Table 4). Most FBXW7 mutations are localized to the substrate-binding WD-repeats [122]. FBXW7 mutation status also correlates with tumor grade, EC type, and lymph node status [123]. Targetable FBXW7 mutations (6%) have also been frequently found in EC [124]. Numerous other studies have reported that FBXW7 mutated with a frequency of approximately 5~30% in ECs [96,125,126,127,128,129,130,131]. Moreover, recurrences primarily align with CIP2A overexpression and PPP2R1A or FBXW7 mutation [132]. Computational analysis shows a significant deviation in structural configuration and stability of FBXW7 mutants R465C, R465H, R465P, R505C, R505G, R505L, and R505S structures. The protein–protein interaction network of FBXW7 consists of hub proteins such as c-Myc, CCNE1, CUL1, KLG5, NFKB2, NOTCH1, SKP1, SREB1, and STYX. Thus, alteration in the FBXW7 leads to aberration in their signaling pathways [133].

In the case of serous EC, higher levels of nuclear FBXW7 and cytoplasmic Protein Phosphatase 2 A, Scaffold Subunit Abeta (PPP2R1B) were associated with a decreased risk of progression [134]. In vitro effects of FBXW7 mutation in serous EC have been shown to increase the sensitivity to SI-2 and dinaciclibraise and raise the levels of potentially druggable proteins [135]. Novel insight into proteomic changes associated with FBXW7 mutation in serous ECs include the identification of PADI2 as a potential therapeutic target for these tumors [136]. FBXW7 mutations have been shown to affect the levels of two druggable proteins: L1 cell adhesion molecule (L1CAM) and transglutaminase 2 (TGM2) [137]. Interestingly, Lehrer and Rheinstein concur with Urick et al. [137] in that L1CAM may be a promising druggable target in EC but did not show any relationship between TGM2 expression, *FBXW7* mutations and EC survival. This suggests that, when compared to L1CAM, TGM2 might not be of as much value as a druggable target [138]. Further studies involving FBXW7, L1CAM, and TGM2 in ECs are needed [139,140]. 

In EC tissues, Liu et al. found that FBXW7 is downregulated, while STYX is upregulated. In addition, STYX interacted with FBXW7 and then downregulated its expression in EC. Over-expression of FBXW7 can also inhibit cell proliferation and facilitate apoptosis in EC cells. Moreover, FBXW7 can suppress the expression of Notch-mTOR pathway-related proteins. Collectively, STYX/FBXW7 axis has been shown to participate in the development of EC via Notch-mTOR signaling pathway [141].

### 6.4. FBXW7 in Rare Gynecological Cancers

FBXW7 has also been shown to be mutated in uterine carcinosarcoma (UCS) and vulvar cancer (Table 5), both of which are rare gynecological malignancies.

UCS accounts for less than 5% of uterine cancers [142]. It is a metaplastic carcinoma which has both sarcomatous and carcinomatous components. The sarcomatous component resembles homologous histologic components, which are typically found in the uterus, or else harbor heterologous components that are not normally present in the uterus, such as rhabdomyosarcomatous or chondrosarcomatous differentiation, and this, by definition, makes it high-grade. The carcinomatous (epithelial) component is also high-grade and typically shows endometrioid or serous differentiation [143]. UCS shares mutational features such as somatic TP53 mutations and extensive copy number alterations, which are more similar to serous uterine carcinoma rather than endometrioid.

According to the conversion theory, UCSs are thought to have a monoclonal origin, where carcinomatous subclones can undergo metaplastic differentiation and transform into sarcomatous cells later on [144]. This theory is backed by the fact that there is the co-expression of cytokeratins and epithelial membrane antigens in carcinomatous and sarcomatous cells, together with a concordance of TP53 and KRAS mutations, identical patterns of X chromosome inactivation, and similar loss of heterozygosity between the carcinomatous and sarcomatous components. Other frequent mutations have been detected in PPP2R1A, PIK3CA, and PTEN, similar to endometrioid and serous uterine carcinomas [144]. The 5-year survival rate is 33–39% [144]. In the case of metastasis, adjuvant treatment largely includes the use of paclitaxel and carboplatin. However, to date, in the case of UCS, there is no trial which has shown an overall survival benefit from adjuvant chemotherapy or radiotherapy [145,146].

Mutations in ARID1A, FBXW7, KMT2C, PIK3CA, KRAS, PTEN, and TP53 play an important role in UCS [148]. However, there are some differences in mutation frequencies, thus reflecting the pathological heterogeneity of UCS. The numerous potential driver genes suggest that this is a genetically heterogeneous disease [147]. Several studies have reported that FBXW7 is frequently mutated in uterine carcinosarcoma (UCS) with a prevalence of over 18% [148,149,150,151,152]. An integrated genomic and proteomic analysis of 57 UCSs has revealed alterations in canonical pathways, particularly in the PI3K pathway, with more than 75% of cases exhibiting a mutation in FBXW7, loss of RB1, or amplification of CCNE1, indicating cell cycle dysregulation [144]. TP53 mutations are commonly acquired in most tumors, indicating that FBXW7, p53, and PI3K pathways play critical roles in UCS, with FBXW7 being a key driver of this specific cancer. Formal genetic evidence from lineage tracing studies suggests that UCS originates from endometrial epithelial cells that undergo an epithelial-mesenchymal transition, leading to a highly invasive phenotype specifically driven by FBXW7 [153].

Vulvar cancer, another rare form of gynecological cancer, has a peak incidence around the age of 80 years but is increasingly being diagnosed at a median age in the fifth decade [157]. Nearly 30% of vulvar cancer cases are diagnosed at FIGO stage III or IV, with a 5-year overall survival of 43% and 13%, respectively [158]. The standard treatment for vulvar cancer is radical surgery with adjuvant radiotherapy in selected cases. To target the most effective treatment while minimizing unnecessary interventions, especially in elderly patients, pathological and clinical prognostic factors are constantly being explored.

Squamous cell carcinoma (SCC) accounts for 80–90% of vulvar carcinomas and is known for its clinical and pathological heterogeneity. Around 43–60% of vulvar SCC are thought to be caused by HPV, leading to the inactivation of TP53 and retinoblastoma (Rb) by E6 and E7 oncoproteins, respectively [159]. These tumors are often associated with diffuse p16 expression [160,161]. The remaining vulvar SCCs are considered “HPV-independent” and tend to occur in older women and are associated with chronic inflammation such as lichen sclerosis [160]. TP53 somatic mutations, PTEN mutation, and EGFR activation have been observed in HPV-independent SCC [159]. Given the rarity and heterogeneity of vulvar SCC, the understanding of the disease’s pathogenesis is limited, and there is a lack of knowledge about targetable molecular pathways. Variants in several genes, including FBXW7 (4%), have been identified in a vulvar cancer cohort using targeted hot-spot sequencing, along with specific protein changes for targetable genes [154]. Thus, molecularly guided precision medicine could offer alternative, targeted treatment options for vulvar cancer patients [154]. In HPV (+) SCCs, novel mutations in BRCA2, FBXW7, and PIK3CA were discovered by Han et al., which had not been previously reported in vulvar SCCs [155]. While HPV (−) SCCs exhibited more nonsilent and driver mutations than HPV (+) SCCs, there was no distinction between HPV (+) and HPV (−) SCCs in terms of CNA loads and mutation signatures. SCCs of vulvar with HPV (+) and HPV (−) may have different mutation and CNA profiles but share common genomic features [155]. Similarly, when screening hrHPV(+) and hrHPV(−) vulvar tumors, CDKN2A and TP53 had common mutations (25% and 21% in hrHPV(+), 46% and 41% in hrHPV(−) cases, respectively) [156]. Other mutations that were identified, although at low frequencies, included FBXW7, AKT1, FLT3, FGFR3, GNAQ, HRAS, JAK3, PIK3CA, PTEN, STK11, and SMAD4. The majority of these mutations could activate the PI3K/AKT/mTOR pathway. Despite being initiated from different premalignant lesions, the genetic mechanisms of the two routes of vulvar SCC pathogenesis may be similar [156].

Overall, these studies suggest that FBXW7 could play a potential role as a prognostic marker and serve as a tumor suppressor for the development of novel and effective therapies for GC patients.

## 7. Potential Therapeutic Strategies Targeting FBXW7

FBXW7 is a protein that plays a critical role in the degradation of specific oncoproteins that promote oncogenesis through the activation of signaling pathways. Reduced FBXW7 expression leads to an accumulation of these oncoproteins and is associated with tumor growth. Strategies that target cancers with FBXW7 loss of function can include the reactivation of FBXW7 expression in tumors with wild-type FBXW7, which may have been inactivated through epigenetic mechanisms, or targeting downstream effectors of FBXW7 inactivation in tumors with gene mutations or deletions [8].

Overexpression of wild-type FBXW7 may reverse drug resistance resulting from FBXW7 loss, and studies have demonstrated the significance of FBXW7 promoter hypermethylation in carcinogenesis [55,162,163]. Decitabine, a DNA methyltransferase inhibitor, has been used to inhibit DNA methylation and promote FBXW7 expression while reducing the expression of the oncoprotein MCL-1 in lung cancer cells [164]. Venetoclax, a Bcl-2 inhibitor, when combined with decitabine, resulted in a strong toxic effect against lung cancer cells [164]. In NSCLC cells with loss of FBXW7, treatment with Entinostat, a class 1 HDAC inhibitor, effectively overcame Taxol resistance and may be beneficial in treating aggressive Taxol-resistant NSCLCs and hematological cancer cells that lack expression of FBXW7 [165,166,167]. However, a disadvantage of these therapies is that they have a global impact on cellular gene expression, which may include repressed oncogenes and other hypermethylated genes when DNA methyltransferase drugs are used [8]. One potential method to restore FBXW7 expression in tumor cells with wild-type FBXW7 mRNA is to target FBXW7-miRNAs. Several promising in vitro and in vivo approaches have been shown, such as small molecule inhibitors of miRNAs (SMIR) or LNA-antimiRs that can disrupt miRNA-target interactions. However, there may be unintended adverse effects of miRNAs since each miRNA can target multiple genes [8].

Low levels of FBXW7 protein can result from the hyperactivation of ERK1/2 signaling or PLK1 or Pin1. Therefore, inhibitors for these pathways can restore FBXW7 expression and/or induce cell cycle arrest or apoptosis. Currently, clinical trials are testing the PLK1 inhibitor onvansertib and the Pin1 inhibitor sulfopin [168,169].

An mTOR inhibitor can reverse the oncogenic effect of an FBXW7 mutation when FBXW7 is inactivated [170]. Downregulation of GSK3β reduces c-Myc T58 phosphorylation, which inhibits FBXW7-mediated c-Myc degradation. This leads to c-Myc accumulation, increasing TRAILR5-induced apoptosis both in vitro and in vivo [171]. This can be used to target cells that overexpress c-Myc in FBXW7-deficient cell lines. Vorinostat, a histone deacetylase (HDAC) inhibitor, induces apoptosis through the upregulation of pro-apoptotic proteins Bim and Noxa and the downregulation of Mcl-1 [49]. FBXW7 mutation in squamous cell carcinoma (SCC) increases the expression of Bim and Mcl-1, leading to resistance to standard chemotherapy while increasing susceptibility to HDAC inhibitors [172]. Therefore, the combination of HDAC inhibitors with BH3-mimetic ABT-737 induces strong synergistic cancer cell death [172]. Research has demonstrated that FBXW7 can inhibit metastasis [173]. In particular, when FBXW7 is knocked out in bone marrow-derived stromal cells of mice, Notch expression increases and leads to an elevation in the expression of C-C motif chemokine ligand 2 (CCL2) [174]. Elevated CCL2 levels attract macrophages and monocytic myeloid-derived suppressor cells to the tumor site, which can facilitate metastasis [173]. However, the depletion of FBXW7 can be reversed using a CCL2 receptor antagonist, which presents a promising therapeutic strategy for targeting FBXW7-deficient cells [174].

Interestingly, FBXW7 plays a crucial role in maintaining stem cells in various tissues, including cancer stem cells (CSCs), which are a small subset of tumor cells with the ability to initiate tumors, and promote tumor spread, recurrence, and chemoresistance [173]. When FBXW7 is ablated in the mouse model of chronic myeloid leukemia, CSCs enter the cell cycle due to the accumulation of c-MYC [175,176]. Consequently, loss of FBXW7 increases the sensitivity of CSCs to conventional treatment with the tyrosine kinase inhibitor imatinib or cytarabine. The combination of FBXW7 ablation and treatment with these drugs has been shown to eliminate CSCs and decrease the rate of relapse in the murine model [175]. FBXW7 expression was found to be upregulated when treating LGR5+ CSC-enriched cell lines of human CRC with oxaliplatin or irinotecan but not in other non-stem-like cell lines [177]. Depletion of FBXW7 in the LGR5+ CSC-enriched cells led to an accumulation of c-MYC and increased cell sensitivity to chemotherapy.

Despite the growing understanding of the regulation and function of FBXW7 in both normal and tumoral cells, several questions remain unanswered. For instance, mutated forms of FBXW7 may have carcinogenic effects, which could present an opportunity to develop targeted therapies. However, certain specific mutations may differentially degrade several FBXW7 targets, unlike the hot spot arginine mutations (R465, R479, and R505) [8]. Understanding the conformational changes that underlie these phenotypes could help design small inhibitors that selectively affect specific downstream signaling pathways and/or alter FBXW7 substrate selection. It is also essential to investigate the contribution of FBXW7 monomers and dimers in homozygous or heterozygous mutations since FBXW7 mutations are typically heterozygotic [11]. Therefore, the effects of targeting FBXW7 on cancer outcomes are complex. While the elimination of FBXW7 can promote the cell cycle and inhibit apoptosis in cycling cancer cells, it can also be effective in eliminating CSCs. Developing chemical inhibitors of FBXW7 and testing their effects, whether in combination with conventional chemotherapy or alone, should help uncover the potential of FBXW7-targeted therapy for cancer management [9].

Gynecologic cancers (GCs) are a diverse set of tumor types with a unique pattern of alterations. Therefore, there is a pressing need to consider changing the design of clinical trials to include all gynecologic oncology patients in biomarker-based, basket-type clinical trials. Despite multiple clinical trials targeting FBXW7-related signaling pathways, only a handful of them include GCs (Table 6) [21,168,169,178,179,180].

## 8. Conclusions 

FBXW7 is a crucial component of the SCF ubiquitin ligase complex and acts as a tumor suppressor by regulating the abundance of various oncogenic proteins. In GC and other cancers, FBXW7 is frequently mutated and is associated with resistance to treatment and poor prognosis. Therefore, the detection of FBXW7 mutation status can be a valuable diagnostic biomarker and plays a critical role in personalized therapy. Since FBXW7 mutations are mostly heterozygotic, it is essential to understand the effects of monomeric and dimeric forms of FBXW7 on mutational status to further comprehend its function. Comprehensive studies are necessary to investigate the complex network of FBXW7, its substrates, and regulators, which will provide a better understanding of GC pathogenesis and possibly discover novel targets for effective treatment. Targeted therapies for tumor cells with FBXW7 mutations may offer individualized treatment options. Future studies will determine whether targeting FBXW7 alone or in combination with downstream or parallel pathways is more beneficial. Complete ablation of FBXW7 expression can desensitize cancer cells and prevent cancer cell death, indicating that some FBXW7 activity is required for effective anti-cancer therapy. Indirect targeting of FBXW7 may decrease drug resistance, enabling current drug therapies to work more efficiently.

## 9. Future Directions

In the future, studies can focus on restoring the functions of FBXW7 by taking advantage of the fact that most tumor cells contain both wild-type and mutated copies, making them heterozygous. As mutated FBXW7 has a dominant negative effect, selective screening and design of small molecules that can interact with the mutation and disrupt dimerization may help to sequester the mutated FBXW7 and free up the wild-type protein to act as a tumor suppressor. Such an approach can have a positive impact on the development of personalized, precise treatments for patients with GCs.

## Figures and Tables

**Figure 1 cells-12-01415-f001:**
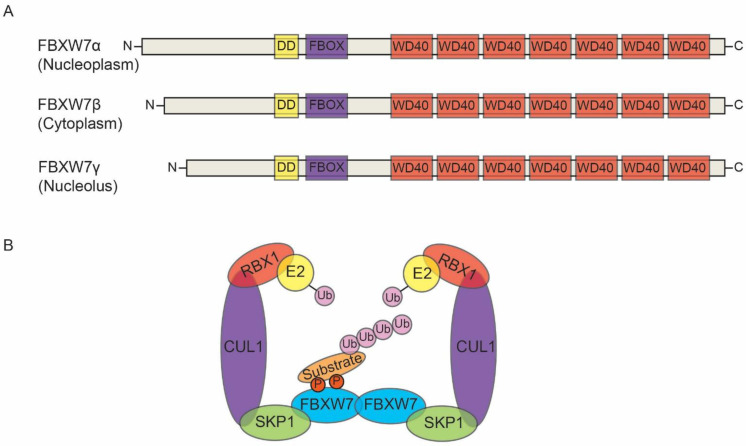
FBXW7 isoforms and SCF-FBXW7 complex. (**A**) Three FBXW7 isoforms (α, β, and γ), which are structurally different only at their N-terminal region, while sharing conserved domains in the C-terminal region. Each of these isoforms consists of three domains: dimerization domain (DD), F-box domain, and tandem WD40 repeats. (**B**) The SCF-FBXW7 complex in FBXW7 dimerization format for substrate ubiquitylation.

**Figure 2 cells-12-01415-f002:**
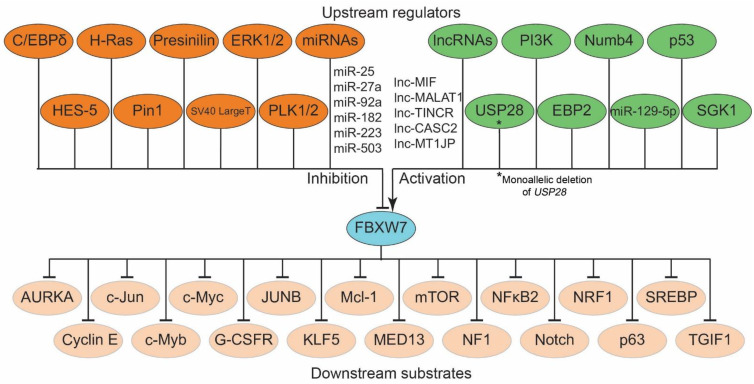
Regulation of FBXW7 showing some of the upstream regulators of FBXW7 and its downstream targets contributing to human tumorigenesis. Several proteins (p53, EBP2, Numb4, SGK1, Pin1, C/EBPδ, HES-5, Presenilin, USP28*, ERK1/2, H-Ras, PI3K, PLK1/2, and SV40 Large T), miRNAs (miR-223, miR-25, miR-27a, miR-182, miR-503, miR-129-5p, and miR-92a), and lncRNAs (MIF, MALAT1, TINCR, CASC2, and MT1JP) regulate the expression of FBXW7. FBXW7 coordinates the ubiquitin-dependent proteolysis of several key oncoproteins (AURKA, Cyclin E, c-Jun, c-Myc, c-Myb, mTOR, KLF5, Mcl-1, NFkB2 SREBP, etc.).

**Figure 3 cells-12-01415-f003:**
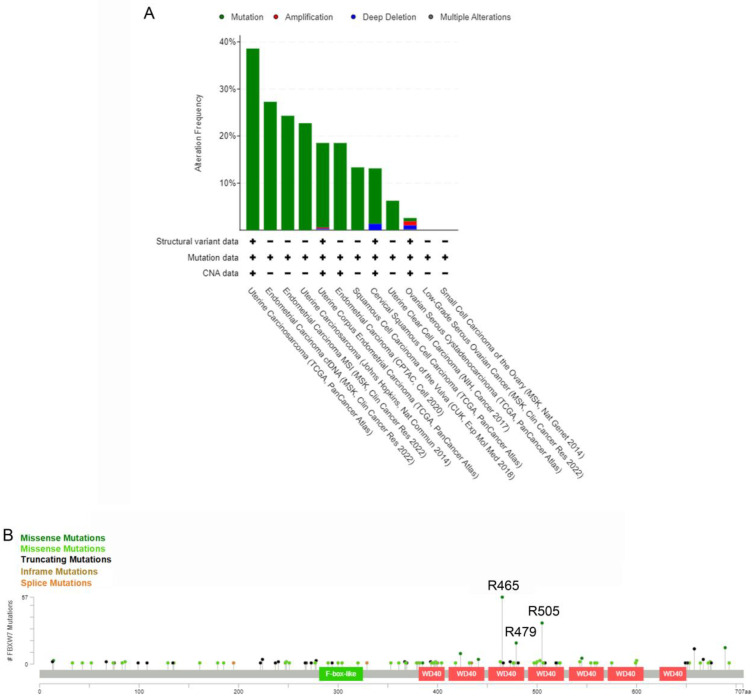
The frequency and distribution of FBXW7 mutation in human gynecological cancers (GC). (**A**) FBXW7 mutation frequency in different types of human GC, analyzed from cBioPortal Database (https://www.cbioportal.org/ (accessed on 30 November 2022)). (**B**) Distribution of FBXW7 mutations in the FBXW7 encoding region. The three most frequent mutation hotspots are R465C/H/L (57/270 mutations), R479Q/G/L/P (18/270 mutations), and R505C/G/H/L (35/270 mutations). **Missense Mutations (*n* = 150)** (putative driver); **Missense Mutations (*n* = 65)** (unknown significance); **Truncating Mutations (*n* = 50)** (putative driver): Nonsense, Nonstop, Frameshift deletion, Frameshift insertion, Splice site; **Inframe Mutations (*n* = 3)** (unknown significance): Inframe deletion, Inframe insertion; and **Splice Mutations (*n* = 2)** (putative driver). Both the figures are from the portal.

**Table 1 cells-12-01415-t001:** Non-coding RNA regulation of FBXW7 in cancer.

ncRNA Type	Role	Cancer Type	Mechanism	FBXW7 Expression	Effect	Sources	Reference
**miRNAs**							
miR-25	Oncogene	NSCLC	FBXW7 is a direct target of miR-25	Downregulated	Promote cell proliferation, migration and invasion	Tissue samples, cell lines	[33]
miR-27a	Oncogene	ALL	miR-27a controls FBW7-dependent cyclin E degradation	Downregulated	Increases DNA replication stress and alters cell cycle progression	Tissue samples, cell lines	[34]
miR-92a	Oncogene	CC	FBXW7 is a direct target of miR-92a	Downregulated	Promote cell proliferation and invasion	Tissue samples, cell lines	[35]
miR-182 and miR-503	Oncogene	Colorectal cancer	miR-182 and miR-503 cooperatively target FBXW7	Downregulated	Increases tumor growth	Tissue samples, xenograft models, cell lines	[37]
miR-223	Oncogene	T-ALL	TAL1 targets FBXW7 through miR-223	Downregulated	Promotes cell growth and inhibits apoptosis	T-ALL samples, T-ALL cell lines	[38]
**lncRNAs**							
lnc-MIF	Tumor suppressor	Different cancer types	lnc-MIF increases FBXW7 expression by acting as a molecular sponge for miR-586	Upregulated	Inhibit aerobic glycolysis and tumorigenesis	Xenograft models, cell lines	[39]
lnc-MT1JP	Tumor suppressor	GC	lnc-MT1JP regulated FBXW7 expression by competitively binding to miR-92a-3p	Upregulated	Inhibit cell proliferation, migration and invasion	Tissue samples, xenograft models, cell lines	[40]
lnc-MALAT1	Tumor suppressor	Glioma	lnc-MALAT1 increases FBXW7 expression by down-regulating miR-155	Upregulated	Inhibit cell viability	Tissue samples, cell line	[41]
lnc-TINCR	Tumor suppressor	Lung cancer	lnc-TINCR increases FBXW7 expression via its action as a molecular sponge for miR-544a	Upregulated	Inhibit cell proliferation, migration and invasion	Tissue samples, cell lines	[42]
lnc-CASC2	Tumor suppressor	HCC	lnc-CASC2 increases FBXW7 expression via its action as a molecular sponge for miR-367	Upregulated	Inhibit EMT	Tissue samples, cell lines	[43]

Abbreviations. ncRNA: Non-coding RNA; NSCLC: non-small cell lung cancer; ALL: acute lymphoblastic leukemia; CC: cervical cancer; T-ALL: T cell acute lymphoblastic leukemia; HCC: hepatocellular carcinoma; GC: gastric cancer; and EMT: epithelial-mesenchymal transition.

**Table 2 cells-12-01415-t002:** Role of FBXW7 in ovarian cancer.

Study	Source	Mutation Type and/or Expression	Protein Change	Modulation of FBXW7 and/or Mechanism of Action	Effect	Clinicopathological Significance
Jardim et al. [21]	Patients	Missense	R465H	N.A.	N.A.	Limited therapeutic efficacy of mTOR inhibitors
Sakai et al. [75]	FFPE tissues	Missense	R505G; R505L	N.A.	N.A.	N.A.
Boyd et al. [76]	Tumor tissues	Nonsense	Q430X	N.A.	N.A.	*FBXW7* and *KIAA1462* genes are candidates for a pathogenic role in SBT
Aziz et al. [77]	Tumor tissues	Low or absent expression	N.A.	N.A.	N.A.	High chromosomal instability and poor patient outcome
Kitade et al. [74]	Tumor tissues; cell line	Downregulated	N.A.	Mutated p53 suppresses *FBXW7* expression	N.A.	No significant difference in overall survival between the high and the low *FBXW7* expression groups
Zhao et al. [78]	Cell line	Downregulated	N.A.	MAGEA1 promotes NICD1 ubiquitination and degradation by promoting the interaction between FBXW7 and NICD1	Inhibits cell proliferation and migration	N.A.
Noack et al. [79]	Cell lines; primary cell cultures	Downregulated	N.A.	PLK1 inhibitor BI6727/paclitaxel-co-treatment stabilizes FBXW7	Induces apoptosis	N.A.
Xu et al. [80]	TCGA data; cell lines; xenograft tumor model	Downregulated	N.A.	*FBXW7γ* overexpression reduces protein expression of c-Myc, Notch1 and Yap1	Inhibits cell growth in vitro and in vivo	*FBXW7γ* expression is an independent indicator of longer disease-specific survival and progression-free survival
Liu et al. [81]	Tumor tissues; cell lines	Downregulated	N.A.	Circ-BNC2 overexpression upregulates FBXW7 via sponging miR-223-3p	Inhibits cell proliferation, migration and invasion	N.A.
Xu et al. [82]	Tumor tissues; cell lines; xenograft tumor model	Downregulated	N.A.	Ectopic FBW7 induces proteasomal degradation of YTHDF2	Inhibits cell survival and proliferation in vitro and in vivo	High expression is associated with favorable prognosis
Miao et al. [83]	Tumor tissues; cell lines	Downregulated	N.A.	TTN-AS1overexpression upregulates FBXW7 via sponging miR-15b-5p	Inhibits cell proliferation, colony formation and promotes apoptosis	N.A.
Guo et al. [84]	Cell lines	Downregulated	N.A.	APS upregulates FBXW7 by miR-27a down-regulation.	Inhibits cell proliferation, migration, invasion and promotes apoptosis	N.A.

Abbreviations. N.A. not assessable; FFPE: formalin-fixed, paraffin-embedded; SBT: Serous borderline tumor; TCGA: The Cancer Genome Atlas; and APS: Astragalus polysaccharide.

**Table 3 cells-12-01415-t003:** Role of FBXW7 in cervical cancer.

Study	Source	Mutation Type and/or Expression	Protein Change	Modulation of FBXW7 and/or Mechanism of Action	Effect	Clinicopathologic Significance
Spaans et al. [96]	FFPE tissues; Tumor tissues	Missense	R465C; R465H	N.A.	N.A.	N.A.
Ojesina et al. [97]	Tumor tissues	Missense	(R465C; R465H; R479P; R505G; R543T) *	N.A.	N.A.	N.A.
Spaans et al. [98]	FFPE tissues	Missense	R465C; R465H; R479L; R479Q	N.A.	N.A.	N.A.
Luo et al. [99]	GEO and TCGA data	CNA	N.A.	N.A.	N.A.	N.A.
Huang et al. [100]	Patients	Different types of genetic alterations	N.A.	N.A.	N.A.	N.A.
Kuno et al. [101]	TCGA datasets	Different types of genetic alterations	N.A.	N.A.	N.A.	Genomic alterations in *FBXW7* were not significantly correlated with progression free survival
Kashofer et al. [102]	Micro-dissected samples	Missense	D399N; R465H; R479Q; R505G	N.A.	N.A.	N.A.
Muller et al. [103]	Tumor tissues	Missense	R465C; R479Q; R505G	N.A.	N.A.	N.A.
Li et al. [104]	Tumor tissues	Missense	R465C; R465H; R505G	N.A.	N.A.	N.A.
Tian et al. [105]	Blood samples	Non-synonymous	*	N.A.	N.A.	MRCC patients with any detectable MSG mutations had significantly shorter progression free and overall survival than those without detectable MSG mutations
Lu et al. [106]	Several TCGA datasets	Mutation and epigenetic silencing	N.A.	N.A.	N.A.	Genomic and epigenetic alterations in *FBXW7* are exhibited in immune subtype of CSCC HPV16-IMM
Liu et al. [107]	Tumor tissues; cell lines	Missense	L443H; R479P	N.A.	Both of these mutations promote cell proliferation, migration, and invasion	N.A.
Wang et al. [108]	TCGA-CESC and GSE44001 datasets	Different types of genetic alterations	N.A.	N.A.	Aberration in several pathways	Prognostic model based on oxidative stress-related genes
Zhou et al. [35]	Tumor tissues; cell lines	Downregulated	N.A.	miR-92a down-regulation upregulated FBXW7	Inhibits cell proliferation and invasion	N.A.
Xu et al. [109]	Tumor tissues	Downregulated	N.A.	N.A.	N.A.	Loss of *FBXW7* is related to poor prognosis
Zhang et al. [110]	TCGA data; Tumor tissues; cell lines	Downregulated	N.A.	LINC00173 overexpression upregulated FBXW7 via sponging miR-182-5p	Inhibits cell proliferation and invasion	N.A.
Ben et al. [111]	Tumor tissues; cell lines	Downregulated	N.A.	miR-27a-3p down-regulation upregulated FBXW7	Inhibits cell proliferation, colony formation and promotes apoptosis	N.A.
Ren et al. [112]	GSE55940 data; cell lines	Downregulated	N.A.	miR-103a-3p down-regulation upregulated FBXW7	Reduces viability by inducing apoptosis	N.A.

Abbreviations. FFPE: formalin-fixed, paraffin-embedded; N.A. not assessable; CNA: copy number alteration; TCGA: The Cancer Genome Atlas; CSCC: Cervical squamous cell carcinoma; MRCC: metastatic relapsed cervical cancer; and MSG: metastatic relapse significantly mutated. * Please see the cited article, for a complete list of mutations.

**Table 4 cells-12-01415-t004:** Role of FBXW7 in endometrial cancer.

Study	Source	Mutation Type and/or Expression	Protein Change	Modulation of FBXW7 and/or Mechanism of Action	Effect	Clinicopathologic Significance
Le Gallo et al. [122]	Tumor tissues	Missense	(R465C; R465H; R479Q; R505C; Y545C) *	N.A.	N.A.	Loss of FBXW7 function may be correlated with resistance to antitubulin chemotherapy and sensitivity to an HDAC inhibitor
Garcia-Dios et al. [123]	Five datasets; FFPE tissues; Tumor tissues	Missense	(R465C; R465H; R479G; R479L; R479Q) *	N.A.	N.A.	*FBXW7* mutants correlated with a positive lymph node status
Stelloo et al. [124]	Tumor tissues	Missense	R465C; R465H	N.A.	N.A.	Somatic mutations in *FBXW7* can potentially be targetable with HDAC inhibitors
Spaans et al. [96]	FFPE tissues; Tumor tissues	Missense	R465C; R465H; R505C	N.A.	N.A.	N.A.
DeLair et al. [125]	TCGA data; FFPE tissues	Different types of genetic alterations	*	N.A.	N.A.	N.A.
Cuevas et al. [126]	TCGA data; Tumor tissues	Missense	R505C; R505G	N.A.	N.A.	N.A.
Lupini et al. [127]	Tumor tissues	Missense	R361Q	N.A.	N.A.	N.A.
Bosquet et al. [128]	TCGA data; cell lines	Different types of genetic alterations	N.A.	N.A.	N.A.	Integration of *CCNA2* and *E2F1* overexpression and *PPP2R1A*, *POLE*, and *FBXW7* mutations generated a molecular EC classification which projects prognostic risk, platinum insensitivity and potential targetable therapeutic options
Feng et al. [129]	CLISING and cBioportal database; tumor tissues; blood samples	Different types of genetic alterations	*	N.A.	N.A.	*TP53*, *PIK3CA*, *PTEN*, *PIK3R1*, and *FBXW7* mutations were not related to FIGO stage or recurrence. However, personalized ctDNA detection for one-to-three high-frequent mutations, was useful in monitoring high-risk EC relapse during post-operative follow-up as a prognostic marker
Ross et al. [130]	Tumor tissues	Different types of genetic alterations	N.A.	N.A.	N.A.	N.A.
Lin et al. [131]	TCGA data; Tumor tissues	Different types of genetic alterations	N.A.	N.A.	N.A.	N.A.
Gonzalez-Bosquet et al. [132]	TCGA data	Different types of genetic alterations	N.A.	N.A.	N.A.	*CIP2A* overexpression or *PPP2R1A*-mut or FBXW7-mut or a combination of these aberrations was negatively associated with progression free survival
Vasuki and Christy [133]	TCGA data	Missense	(R465H; R465P; R505C; R505G; R505L) *	Interaction with NOTCH1, c-Myc, CCNE1, STYX, KLG5, SREB1, NFKB2, SKP1 and CUL1	Aberration in their signalling pathways	N.A.
Dinoi et al. [134]	TCGA data; Tumor tissues	N.A.	N.A.	N.A.	N.A.	Higher nuclear FBXW7 and cytoplasmic PPP2R1B levels were associated with a decreased risk of progression
Urick et al. [135]	Cell lines	Missense	(G423V; R465C; R465H; R479Q; R505C) *	*FBXW7* mutations lead to increased Cyclin E1, SRC-3, c-MYC, Rictor, GSK3, P70S6 kinase and AKT phosphorylated protein levels	FBXW7-mutant cells exhibit increased sensitivity to SI-2 and dinaciclib	N.A.
Urick et al. [136]	Cell lines	Missense	R465C, R479Q; R505C	FBXW7 mutation (R505C) leads to increased PADI2 expression	N.A.	N.A.
Urick et al. [137]	Cell lines	Missense	R465C, R479Q; R505C	FBXW7 mutations increased L1CAM and TGM2 protein levels	N.A.	N.A.
Lehrer and Rheinstein [138]	TCGA data	Different types of genetic alterations	N.A.	FBXW7 mutations affect gene expression of L1CAM but are unrelated to TGM2 gene expression	N.A.	FBXW7 mutations are unrelated to survival
Liu et al. [141]	Tumor tissues; cell lines	Downregulated	N.A.	STYX suppresses FBXW7 expression via direct protein–protein interaction	Over-expression of *FBXW7* inhibits cell proliferation and promotes apoptosis	N.A.

Abbreviations. N.A. not assessable; FFPE: formalin-fixed, paraffin-embedded; CNA: copy number alteration; N.D. not determined; TCGA: The Cancer Genome Atlas; HDAC; Histone deacetylase; SRC-3; steroid receptor coactivator 3; GSK3; glycogen synthase kinase 3; PADI2: peptidyl arginine deiminase 2; and ctDNA; circulating tumor DNA. * Please see the cited article, for a complete list of mutations.

**Table 5 cells-12-01415-t005:** Role of FBXW7 in rare gynecological cancers.

Study	Cancer Type	Source	Mutation Type and/or Expression	Protein Change
Hembree et al. [147]	UCS	COSMIC and TCGA data from patients	Missense; nonsense	R465H; R658X
McConechy et al. [148]	UCS	FFPE tissues; tumor tissues	Missense	R385C; R387L; R425L; R465C; R505L) *
Le Gallo et al. [149]	UCS	Tumor tissues	Missense	(G437V; R465H; R465L; R479Q; Y545C) *
Crane et al. [150]	UCS	Tumor tissues	Different types of genetic alterations	N.A.
Moukarzel et al. [151]	UCS	TCGA data; FFPE tissues; tumor tissues	Missense	R465C
Ashley et al. [152]	UCS	TCGA data	Missense	G423V
Cherniack et al. [144]	UCS	TCGA data; tumor tissues	Missense	(G423V; R465H; R479Q; R689W; S558F) *
Cuevas et al. [153]	UCS	TCGA data; mice models; cell lines	Different types of genetic alterations	*
Palisoul et al. [154]	Vulvar	Database; patients	Frameshift	E471fs
Han et al. [155]	Vulvar	COSMIC and TCGA data; FFPE tissues; tumor tissues	Missense	R399Q ; R463G
Zięba et al. [156]	Vulvar	FFPE tissues; tumor tissues; cell lines	Missense	S462Y, R479G, T482A, R505C, R505G

Abbreviations. UCS: Uterine carcinosarcoma; N.A. not assessable; FFPE: formalin-fixed, paraffin-embedded; TCGA: The Cancer Genome Atlas; and COSMIC: Catalogue of Somatic Mutations in Cancer. * Please see the cited article, for a complete list of mutations.

**Table 6 cells-12-01415-t006:** Summary of some clinical trials targeting FBXW7-related signaling pathways.

Compound	Phase	Malignancy	Target	Trial Registration	Reference
Sirolimus, HCQ	I	Bladder and Colorectal cancers	mTOR	Clinical Trials Program at Department of Investigational Cancer Therapeutics, University of Texas MD Anderson Cancer Center	[21]
Everolimus, pazopanib		Colorectal cancer			
Everolimus, anakinra		Colorectal cancer			
Sirolimus, vorinostat		HCC			
Temsirolimus, bevacizumab, valproic acid		Colorectal cancer			
Sirolimus, lapatinib		Mesothelioma			
Everolimus, Anastrozole		Ovarian cancer			
Onvansertib in combination with either LDAC or decitabine	Ib	AML	PLK1	NCT03303339	[168]
Sulfopin	Pre-clinical	Neuroblastoma and pancreatic mouse model, and neuroblastoma zebrafish model	PIN 1		[169]
Simeprevir + pegylated interferon-α + ribavirin	II/III	Chronic hepatitis C virus	Target FBXW7-miRNAs	NCT01349465	[178]
MLN8237/Alisertib	I/II/III	Relapsed/Refractory peripheral T-cell lymphoma, non-Hodgkin lymphoma, advanced-non hematological malignancies, lung, breast, head and neck, gastroesophageal malignancies, and advanced or metastatic sarcoma	AURKA	NCT01482962 NCT00807495 NCT01045421 NCT01653028	[179]
BI6727 (Volasertib)	II	Ovarian cancer	PLK1	NCT01121406	[179]
Danusertib (PHA-739358)	I	Bcr-Abl-associated advanced hematologic malignancies	AURKA	European Clinical Trails Data Base (EudraCT number 2007-004070-18)	[180]

Abbreviations. HCC: hepatocellular carcinoma; AML: acute myeloid leukemia; LDAC: Low Dose Cytarabine.

## Data Availability

This review paper does not report any new data.

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
