# Peer review of "The Role of FBXW7 in Gynecologic Malignancies"

_cells, 2023, doi:10.3390/cells12101415_

Round 1
Reviewer 1 Report
this is an interesting paper on the role of FBXW7, a protein that regulates cellular growth also in tumoral cells, in gynaecologic malignancies.
The authors are of well-recognized leader groups about this topic. The review is well structurated. Of particular interest are the paragraphs about post-translational regulation of FBXW7 and about genetic and epigenetic alterations causing FBXW7-inactivation.
I do not have any particular concern about this paper and I can only recommend its publication.
Author Response
Dear Reviewer 1, many thanks for the kind comments made on our manuscript.
Reviewer 2 Report
The current study explored the role of FBXW7 in gynecological cancers. The overall study is well-designed and presented. I have suggestions for further improvements.
The authors have mentioned different regulators of FBXW7 including miRNAs and lncRNAs. It would be better if the authors may provide more details in a table for readers' convenience, like each lncRNA, miRNA or others, and their reported functions/pathways, etc.
Better add information about interacting proteins of FBXW7 and interacting RNAs.
Please provide the expression status of FBXW7 in different gynocological cancers.
Tables, Please mention the type of protein change. expression, how, etc?
Are there any clinical trials of FBXW7 or its targets? Please highlight
Please provide the key questions which require scientific attention.
Please write the conclusion and prospects separately.
Author Response
Dear Reviewer 2, many thanks for the kind comments made on our manuscript and for the useful suggestions.
Answer 1. A new table, showing the mechanisms of regulation of FBXW7 by miRNAs and lnc-RNAs, is reported (Table 1).
Answer 2. In Table 1, information about nc-RNAs (miRNAs and lnc-RNAs), about their roles, mechanism of action and certain effects in certain tumor types are summarized.
Answer 3. In the tables, in the “Mutation type and/or expression” column, the types of mutation and expression status (up-regulated or down-regulated) of FBXW7 are reported. Where only mutational status or expression status is reported, it is due to a lack of such information in the cited article.
Answer 4. The tables have been revised. The information relating to Mutation type and/or expression, Protein change, effects and Clinicopathological significance of FBXW7, where it was possible to obtain from the articles, are reported
Answer 5. Although multiple clinical trials are currently targeting FBXW7-related signaling pathways, only a few of them include GCs. A new table is reported (Table 6). Moreover, the following references have been added:
- Zoulim, F.; Moreno, C.; Lee, S.S.; Buggisch, P.; Horban, A.; Lawitz, E.; Corbett, C.; Lenz, O.; Fevery, B.; Verbinnen, T.; et al. A 3-year follow-up study after treatment with simeprevir in combination with pegylated interferon-α and ribavirin for chronic hepatitis C virus infection. Virol J. 2018, 15,26.
- Llombart, V.; Mansour, M.R. Therapeutic targeting of "undruggable" MYC. EBioMedicine 2022, 75,103756.
-Borthakur, G.; Dombret, H.; Schafhausen, P.; Brummendorf, T.H.; Boissel, N.; Jabbour, E.; Mariani, M.; Capolongo, L.; Carpinelli, P.; Davite, C.; et al. A phase I study of danusertib (PHA-739358) in adult patients with accelerated or blastic phase chronic myeloid leukemia and Philadelphia chromosome-positive acute lymphoblastic leukemia resistant or intolerant to imatinib and/or other second generation c-ABL therapy. Haematologica 2015, 100,898-904.
Answer 6. In the section "Potential Therapeutic Strategies Targeting FBXW7" the limits of the various therapeutic strategies that provide for the reactivation or restore of FBXW7 are highlighted. So, a better understanding of the molecular mechanisms is necessary to set up a specific and effective therapy. Furthermore, as always described in the same section, it is extremely important to understand the role it plays in the regulation of stemness, as well as the function of the mutated forms of FBXW7 which could acquire oncogenic functions. These issues that need to be clarified by future studies are also emphasized in the Conclusions and Future Directions sections.
Answer 7. Conclusions and Future Directions have been reported in separate sections
Reviewer 3 Report
The manuscript is well presented and comphensive. The authors focus their manuscript on the role of FBXW7 in Gynecologycal Cancers. After having explained the structure, the regulation and the wide function of FBXW7, the authors reported the evidences of its involvement in GC. These informations are useful either for potential biomarker sas well as to individuate potential therapeutical target. The tables and the figures are appropriate.
Author Response
Dear Reviewer 3, Thank you for your kind comments.
Round 2
Reviewer 2 Report
The authors have addressed all of my comments
Author Response
The repetition rate has been further reduced.
Thanks